

# BUMPER v1.0: A Bayesian User-friendly Model for Palaeo-Environmental Reconstruction

Philip B. Holden[1], H. John B. Birks[2], Stephen J. Brooks[3], Mark B. Bush[4], Grace M. Hwang[5], Frazer Matthews-Bird[4], Bryan G. Valencia[4] and Robert van Woesik[4]

[1]Earth, Environment and Ecosystems, The Open University, Walton Hall, Milton Keynes MK7 6AA, UK
[2]Department of Biology, Universty of Bergen, PO Box 7803, N-5020 Bergen, Norway; and Environmental Change Research
Centre, University College London, London WC1E 6BT, UK
[3]Department of Entomology, Natural History Museum, Cromwell Road, London SW7 5BD, UK
[4]Department of Biological Sciences, Florida Institute of Technology, 150 West University Boulevard, Melbourne, FL 32901, USA
[5]The Johns Hopkins University Applied Physics Laboratory, 11000 Johns Hopkins Road, Laurel, MD 20723, USA

*Correspondence to*: Philip B. Holden (philip.holden@open.ac.uk)

**Abstract.** We describe the **B**ayesian **U**ser-friendly **M**odel for **P**alaeo-**E**nvironmental **R**econstruction (BUMPER), a Bayesian transfer function for inferring past climate from microfossil assemblages. BUMPER is fully self-calibrating, straightforward to apply, and computationally fast, requiring ~2 seconds to build a 100-species model from a 100-site training-set on a
standard personal computer. We apply the model's probabilistic framework to generate thousands of artificial training-sets under ideal assumptions. We then use these data to demonstrate the sensitivity of reconstructions to the characteristics of the training-set, considering assemblage richness, species tolerances, and the number of training sites. We find that a useful guideline for the size of a training-set is to provide, on average, at least ten samples of each species. We demonstrate general applicability to real data, considering three different organism types (chironomids, diatoms, pollen) and different
reconstructed variables. An identically configured model is used in each application, the only change being the input files that provide the training-set environment and species-count data. The performance of BUMPER is shown to be comparable with Weighted Average Partial Least Squares (WAPLS) in each case. Additional artificial datasets are constructed with similar characteristics to the real data, and these are used to explore the reasons for the differing performances of the different training-sets.


## 1 Introduction

Transfer functions are numerical tools that are widely used to infer past environment from species assemblages of microfossils preserved in lake or ocean sediments (Imbrie and Kipp, 1971; Birks and Seppä, 2004; Birks et al., 2010). These
species assemblages can provide a strong indicator of the environmental characteristics of the habitat in which the microfossils were found. Numerical relationships can be developed between present-day species distributions and the environment, and by applying the principle of uniformitarianism (Rymer, 1978), these relationships can estimate palaeoenvironments by examining the presence and abundance of palaeo-species. Here we take a Bayesian approach and develop a transfer-function model that is tested on three different organism types (chironomids, diatoms, pollen). Although
Bayesian transfer functions have a moderately long history (Vasko et al., 2000; Korhola et al., 2002; Haslett et al., 2006) they are still not routinely employed by palaeoecologists, who generally apply frequentist approaches, such as weighted-averaging (Birks et al., 1990). The advantages of frequentist approaches are that they are relatively straightforward to understand and apply, and that they are computationally fast. In contrast, Bayesian approaches tend to involve more complex mathematics and can be computationally demanding. The advantages of Bayesian approaches, however, are numerous and include the use
of prior information (van Woesik, 2013).



The principal motivation for a Bayesian approach is that the palaeoenvironment is treated probabilistically, and can be updated as additional data become available. Bayesian approaches therefore provide a reconstruction-specific quantification of the uncertainty in the data and in the model. Bayesian models are in principle ideally suited to multi-proxy reconstructions

because the posteriors that are derived from independent proxies can be combined. However, to our knowledge, no Bayesian approach has yet been applied to a multi-proxy reconstruction, at least in part because of the aforementioned difficulties with application.

Here we describe the "Bayesian User-friendly Model for Palaeo-Environmental Reconstruction" BUMPER and demonstrate

its general validity and ease of use through applications to both artificial and real datasets. The model can be regarded as a Bayesian analogue to weighted averaging-type approaches (Birks et al., 1990) because it assumes a unimodal response of each taxon to the reconstructed environmental variable of interest. It therefore shares many of the weaknesses of weighted averaging-type approaches (Huntley, 2012; Juggins, 2013), most notably by assuming that the responses to different environmental variables are independent. It is the task of the ecologist, assisted by multivariate techniques such as ordination

(Hill and Gauch, 1990; Juggins and Birks, 2012; Juggins 2013), to ascertain which environmental variable is the dominant driver of change in the assemblage through time. When strong interactions are suspected, approaches that concurrently reconstruct the interacting variables are advised (Huntley et al., 1993; Huntley, 2012).

The mathematics of BUMPER was developed and applied to the Surface Water Acidification Programme (SWAP) diatom

training-set (Stevenson et al., 1991) in Holden et al. (2008). The mathematics is unchanged here. Instead the primary motivation of the present study is to document the developments to the model needed to make it user-friendly. The two important steps were first to develop an approach to generalise and automate the priors for the model parameters (we do not assume that the user has the necessary expertise to make these decisions), and second to demonstrate that the model can be applied to other organism types and environmental variables. We apply the model to both artificial and real data, considering

a range of different organism types to demonstrate the model's general applicability.

## 2 The model

The mathematics of BUMPER has been described in detail (Holden et al., 2008). In this section we summarize the

underlying philosophy. The principal assumption underlying the model is that biological species exhibit a unimodal response to an environmental variable, so that both the probability of presence and the expected abundance of a given species are maximized at the environmental optimum favored by that species. Moreover, the species optima are fixed, so that the response to the environmental variable of interest is assumed to be independent of other environmental variables.

### 2.1 Model selection


A species response curve (SRC) defines the probability of an observed count of a species, or taxon, as a function of the environmental variable of interest. An SRC is defined by five (initially unknown) parameters. These parameters specify the environmental optimum and tolerance of each taxon, together with the probability of its presence, and its expected

abundance under optimal conditions.





BUMPER considers 2,560 possible SRCs for each taxon[1] and applies Bayesian model selection to quantify how well each of the SRCs fits the observations of the training-set. Bayes equation states that the probability that a model (each of the SRCs) is correct, in light of observational data, is proportional to the probability that the data would be observed if the model were correct. Expressed more formally, for each of the $j=1,2560$ SRCs considered for each taxon $k$, probability weights are derived from the observed environmental variable $x_i$ and the species count $y_{ik}$ in each training-set site $i$:

$$prob(SRC_{jk}|y_{ik},x_i) \propto prob(y_{ik}|SRC_{jk},x_i) \times prob(SRC_{jk}) \tag{1}$$

The second term on the right hand side is the prior, the probability that the model is correct before we consider the data. Equation 1 is applied sequentially across all training-set sites. At each application, the posterior derived from the previous training-set site becomes the prior for application to the next site, so that the probabilities assigned to each SRC become progressively better defined. The proportionality constant means that we have to normalise the calculated probabilities after completing this process for all SRCs. This is achieved from the assumed constraint that for each species $k$, as follows:

$$\sum_j prob(SRC_{j,k}) = 1 \tag{2}$$

### 2.2 Environmental reconstruction

To reconstruct the environment from an observed fossil count $y_{k0}$ of each taxon $k$ within the fossil assemblage, the probability-weighted SRCs are used to derive likelihood functions for each taxon of the reconstructed variable $x$. Here we again use the Bayes relationship, this time stating that the probability that a reconstructed value is correct in the light of an observed species count is proportional to the probability that the species count would be observed in that environment. As we do not know the true SRC with certainty (in general, the calibration will have resulted in many SRCs with non-zero probability), we perform this calculation for all significant SRCs[2] and combine them into a single likelihood function using the probability weights calculated in Section 2.1, as follows:

$$L_y(x|y_{k0}) \propto \sum_j \{prob(SRC_{jk}) \times prob(y_{k0}|SRC_{jk},x)\} \tag{3}$$

An alternative and less constrained presence-absence likelihood function can also be derived as follows:

$$L_p(x|y_{k0}) \propto \sum_j \{prob(SRC_{jk}) \times prob(presence_{k0}|SRC_{jk},x)\} \tag{4}$$

The likelihood function assumed by BUMPER is derived as a linear combination of these two functions, as follows:

$$L(x|y_{k0}) = (1-\eta)L_y(x|y_{k0}) + \eta L_p(x|y_{k0}) \tag{5}$$

The motivation for this blended likelihood function is that $L_y$ provides a tighter constraint on the posterior, but the wide tails contributed by $L_p$ reduce the possibility of the correct solution being ruled out by an outlying species, for instance resulting from misidentification or unusual taphonomic processes. We assume $\eta = 0.5$, except when building a purely binary, presence/absence model, when $\eta = 1.0$.

---

[1] The model of Holden et al. (2008) considered 8,000 SRCs. We here reduce the SRC resolution from (20, 4, 5, 5, 4) to (10,
[2] For computational speed, SRCs are only considered in the reconstruction if they have a probability greater that 1% of the probability of the most probable SRC of that taxon.



The posterior probability distribution for the reconstructed variable $prob(x)$ is derived by combining any prior knowledge $prob(x')$ with the assemblage of likelihood functions, as follows:

$$prob(x) \propto prob(x') \times \prod_k L(x|y_{k0}) \tag{6}$$

A point estimate for the reconstructed variable is derived as:

$$\hat{x} = \int x \, prob(x) \, dx \tag{7}$$

A simple uncertainty metric for the reconstruction is derived as:

$$\Delta = \sqrt{\int (x - \hat{x})^2 \, prob(x) \, dx} \tag{8}$$

**2.3 Probability distribution**

To apply Bayes' equation we require an expression for the probability of observing some count $y$ as a function of the environmental variable $x$. The assumption of a unimodal response leads us to assume that the expected abundance (given presence) $N_{ik}$ of species $k$ in site $i$ follows a Gaussian distribution about some optimum value $u_k$ of the environmental

variable $x_i$, written as:

$$N_{ik} = N_k e^{-(x_i - u_k)^2 / 2t_k^2} \tag{9}$$

Here $N_k$ is the expected abundance (given presence) at the species optimum, and the tolerance $t_k$ is a measure of how rapidly

the expected abundance falls off away from this optimum.

The probability of presence (e.g. used to calculate the $L_p$ likelihood function, Equation 4) is also assumed to follow a Gaussian distribution about the same optimum, though not necessarily of the same tolerance, and can be written in terms of the expected abundance, as:


$$p_{ik} = p_k (N_{ik}/N_k)^{P_k} \tag{10}$$

where $P_k$ is needed because we do not require the same tolerance for $N_k$ and $p_k$, although we can achieve this with $P_k = 1$. We illustrate the role of $P_k$ through example; $P_k \ll 1$ describes a species that is found at high abundance only in an

environment that is relatively close to its environmental optimum, but may also be present (albeit only at low abundance) far away from this optimum.

The probability of a non-zero count $y_{ik}$ (used to calculate the $L_y$ likelihood function, Equation 3) is assumed to follow an exponential decay, with the decay constant defined by the expected count $= 1/N_{ik}$, normalized so that the total probability of

all (non-zero) percentage counts equals the probability of presence $p_{ik}$, and is expressed as:

$$prob(y_{ik}|x_i) = \frac{(p_{ik}/N_{ik}) exp(-y_{ik}/N_{ik})}{1 - exp(-100/N_{ik})} \tag{11}$$





The denominator normalizes the integral over the range $0 < y_{ik} \leq 100$. We note that the denominator is missing from the description of Holden et al. (2008), although it was correctly implemented in the model itself. The probability of absence ($y_{ik} = 0$) is given by $1 - p_{ik}$.

**2.4 Species Response Curve (SRC) priors**

The SRCs are defined by five parameters: species optimum $u_k$, tolerance $t_k$, tolerance scaling $P_k$, probability of presence at environment optimum $p_k$, and expected abundance (given presence) at environmental optimum $N_k$. In general we have little a priori knowledge of appropriate SRC values, and require their priors to be uninformative. However, for computational
tractability, we need to eliminate implausible parameter values.

Previous applications (Holden et al., 2008; Matthews-Bird et al., 2016) used uniform priors for the SRC parameters over ranges that were chosen largely on the basis of subjective judgment (and ascribing a probability of zero outside of those ranges). However, requiring subjective judgement is not desirable, given our desire to maximize the user-friendliness of the
model, and it can be rather time consuming. To prevent subjectivity, BUMPER introduces a simple automated process, based upon an indicative species tolerance $t'$ that is a characteristic of the training-set:

$$t' = \frac{1}{m}\sum_k \sqrt{\frac{1}{n_k}\sum_i (x_i - u_k^{WA})^2} \qquad (12)$$

For each species $k$ we consider the environmental variable $x_i$ at each site where the species is present, and derive a Root Mean Square (RMS) distance from the weighted-average optima, $u_k^{WA}$. We average across the $m$ species that have at least two training-set observations ($n_k \geq 2$). The weighted-average optima are defined in the usual way:

$$u_k^{WA} = \sum_i y_{ik} x_i / \sum_i y_{ik} \qquad (13)$$


This metric $t'$ is used to automate the definition of plausible ranges for species optima and tolerance. The approach we take is analogous to precalibrating the parameters of a physical model (Edwards et al., 2011). We only attempt to rule out implausible parameter values and we ascribe equal probabilities to all plausible parameter combinations. The precalibration only seeks to eliminate SRCs with a probability close to zero, and is thereby designed to avoid the trap of over-constraining
the posterior by using the training-set data twice.

We consider ten possible values for the species optima, and four possible values for each of the four other parameters, giving a total of 2560 SRC combinations. The priors for the five SRC parameters are uniform within specified ranges:

1) Species optima $u_k$ are allowed to take values in the range $(x_{min} - t')$ to $(x_{max} + t')$, where $x_{max}$ and $x_{min}$ are the extremes of the training-set environments. We allow species optima beyond the sampled environment range, but optima beyond this extended range are considered unlikely. These ranges are approximately consistent with the subjective priors used in Holden et al. (2008), being training-set $\pm 0.5$ pH units ($t' = 0.63$ pH units), and Matthews-Bird et al. (2016), being training-set $\pm 5°C$ ($t' = 3.0°C$).


2) Species tolerances $t_k$ are allowed to take values in the range $2t'/3$ to $3t'$. Tolerances at the upper limit are presumably unlikely, and in any event would only weakly constrain the reconstruction. Although tolerances below the lower limit may be reasonable, this choice represents a conservative constraint that, for instance, prevents





unrealistically narrow tolerances when a species is poorly sampled within a training-set. These ranges are consistent with the subjective choices in Holden et al. (2008), being 0.4 to 1.7 pH units, equivalent to $0.63t'$ to $2.7t'$, and Matthews-Bird et al. (2016), being 2 to $10°C$, equivalent to $0.67t'$ to $3.3t'$.

3)   The probability of presence at the environmental optimum $p_k$ is allowed to take values in the range $p'_k$ to $2.5p'_k$, where $p'_k$ is the percentage of sites in which the species is present. A value below $p'_k$ is clearly unlikely as the probability of presence at the optimum must be at least as high as the probability of presence across the training-set. The upper limit is less straightforward to define; this choice is discussed further below.

4)   The expected abundance (given presence) at the environmental optimum $N_k$ is allowed to take values in the range $N'_k$ to $2.5N'_k$, where $N'_k$ is the average count of all non-zero observations across the training-set. A value below $N'_k$ is clearly unlikely as the expected abundance at the optimum must be at least as high as the average abundance across the training-set. Again, the choice of upper limit is discussed below.

5)   The tolerance scaling $P_k$ (Eq. 7) is allowed to take values in the range 0.2 to 1.0. Low values ($P_k \ll 1$) are required to represent species that need near-optimum conditions to flourish at high abundance, but can survive even when conditions are far removed from this optimum. Values $P_k > 1$ can be ruled out because this would (nonsensically) describe a species that can flourish at high abundance away from its optimum, but cannot survive there. In Holden et al. (2008), $P_k$ was allowed to take values in the range 0.4 to 1.0. Here we reduce the lower limit to 0.2, for 20     reasons discussed below.

While aspects of the motivation for ranges 3-5 are discussed above, other aspects are not clearly defined by logical considerations, these being the upper limits for $p_k$ and $N_k$, and the lower limit for $P_k$. These ranges were selected empirically on the basis of application of the model to three training-sets, being chironomid-based temperature (Matthews-Bird et al., 25   2016), diatom-based pH (Stevenson et al., 1991) and pollen-based temperature (Bush et al., in prep). We define the posterior value for a parameter to be the probability-weighted value of that parameter across the 2,560 SRCs of the respective taxon. The posterior values of $p_k$, $N_k$ and $P_k$ for each taxon were sorted from smallest to largest and are plotted sequentially in Figure 1, for each of the three training-sets. The horizontal grid lines represent the four values that each parameter is allowed to take. (Note that although the SRC parameters can only take one of four discrete values, the probability-weighted average 30   can take any value within the prior range). It is apparent that the full range of allowed values is required to describe the species responses of all taxa in all three training-sets (i.e., because the extremes of the probability-weighted averages approach the extremes of the allowed parameter range). Furthermore it is apparent that a wider range is not required, although there is some suggestion that a slightly higher upper limit for $p_k$ could have been allowed for the SWAP data (evidenced by the fact that seven of the 225 species take precisely the upper limit $p_k = 2.5p'_k$).


### 3 Artificial training-set data

The probabilistic framework of BUMPER is well suited to the generation of artificial training-sets. In part, our motivation for artificial training-sets is to investigate how the characteristics of the assemblage affect the performance of the transfer 40   function. Additionally, an important motivation is to apply BUMPER to a selection of training-sets with different characteristics to demonstrate that the model can be applied to an arbitrary training-set without tuning or user modification. In all of the analyses that follows (in both Sections 3 and 4) the identical model is applied (although we consider the




sensitivity of the model to two important assumptions). This model internally generates precalibrated priors from the characteristic tolerance $t'$ and the environmental gradient as described in Section 2.4.

The artificial data are generated from the BUMPER probabilistic framework. As a result of generation, the data are idealised and are expected to over-state performance relative to a training-set of real data with otherwise similar characteristics. Consider, for instance on the one hand, that by applying the probabilistic framework to generate the training-set, species responses are forcibly defined to be unimodal and with optima that are independent of other environmental variables. Furthermore, there are no misidentifications, nor any possibility of taphonomic error. On the other hand, it is worth noting that the artificial training-sites are statistically independent; this eliminates potentially over-optimistic performance statistics that can arise from pseudo-replication in real spatial data that have similar assemblage composition because of geography proximity, and not necessarily because they experience the same environment (Telford and Birks, 2005).

### 3.1 Generating artificial training-sets

We generate an artificial training-set for an arbitrary environmental variable, under the following assumptions.

  i) The training-set sites have an observed environment that is randomly sampled from a uniform distribution over some range $E_L$ to $E_H$. This choice is arbitrary, we select $E_L = 100$, $E_H = 200$. The SRC parameters and performance statistics that follow are expressed as a percentage of the environmental gradient $E_H - E_L = 100$.

  ii) The taxa have optima $u_k$ that are randomly sampled from a uniform distribution in the range $u_L$ to $u_H$. Although BUMPER allows optima beyond the sampled gradient, we here simply assume the range of taxa optima is equal to the environmental gradient ($u_L = E_L$, $u_H = E_H$).

  iii) The taxa have tolerances $t_k$ that are randomly sampled from a uniform distribution in the range $t_L$ to $t_H$. The characteristic tolerance of a training-set is likely to impact greatly upon the performance of the transfer function because it drives the species-turnover rate across the environmental gradient.

  iv) The taxa have probabilities-of-presence-at-optimum $p_k$ that are randomly sampled from a modified exponential with scale parameter $\beta_p$, defined for $0 \le p \le 1$.

$$f(p, \beta_p) = \frac{(1/\beta_p) exp(-p/\beta_p)}{1 - exp(-1/\beta_p)} \tag{14}$$

  The denominator adjusts the exponential so that the cumulative probability is 1 at $p = 1$. The scale parameter $\beta_p$ is used to tune the assemblage for species richness.

  v) The taxa's tolerance scaling $P_k$ (Eq. 7) are randomly sampled from a uniform distribution in the range 0.2 to 1.0.

The values of the abundance parameter $N_k$ are drawn from a modified exponential distribution (c.f. Equation 14) with scale parameter $\beta_N$. The value of $\beta_N$ is not an input because an arbitrary choice will lead to an expected total number of counts (i.e., the sum of all species counts within a site) that differs from 100%. An appropriate value of $\beta_N$ is derived from a Monte-Carlo approach; the expected total count is derived from an exploratory 10,000-site training-set with $\beta_N = 10\%$ and then $\beta_N$ is scaled appropriately and the scaled value applied to generate the actual training-set. BUMPER does not model a multinomial probability distribution so that percentage counts generated under this approach are not constrained to sum precisely to 100%, even when the appropriate value of $\beta_N$ is chosen. A pragmatic approach is taken to address this issue by randomly drawing assemblages, accepting only those with a total count of between 90% and 110% until the required number of sites has been generated. In accepted sites the counts are scaled to total 100%.





### 3.2 Generating transfer functions from the artificial data

In this section, we cross-validate transfer functions derived from artificial training-sets with different characteristics to
investigate how these characteristics affect the reconstruction performance. The characteristics we consider are assemblage
richness and assemblage tolerance. We define the characteristic assemblage richness as the average number of species found
in the training-set sites. This is varied in the artificial ensemble through the exponential scale parameter $\beta_p$. A low value of
$\beta_p$ ascribes low probabilities of presence, even under optimal conditions, and leads to species-poor assemblages. We define
the characteristic assemblage-tolerance to be the indicative tolerance $t'$ (Section 2.4). This is adjusted through the prior
range for tolerance, $t_L$ to $t_H$. For each assemblage type considered, training-sets are generated from 25, 100, 250, and 1000
sites, and are comprised of the same 100 randomly generated artificial species.

We consider three assemblages types.

1)  Low richness, high tolerance; $\beta_p = 10\%, t_L = 15\%, t_H = 25\%$, (solved $\beta_N = 40\%$). The assemblage richness that
15       results from these assumptions is 5.7 i.e., the training-set sites contain on average 5.7 species. Tolerances are
typically 20% of the environmental gradient.

2)  High richness, high tolerance; $\beta_p = 50\%, t_L = 15\%, t_H = 25\%$, (solved $\beta_N = 9\%$). The prior for tolerance is
unchanged from (1), but the expected probability of presence at species optima is increased, generating species-rich
assemblages of average richness 18.3.

3)  Low richness, low tolerance; $\beta_p = 20\%, t_L = 5\%, t_H = 15\%$, (solved $\beta_N = 30\%$). The tolerance is halved relative
to (1) and the expected probability of presence at the species optima is approximately doubled. These two changes
together result in a species richness of 6.4, similar to that of (1), but now the assemblage is comprised of species that
are more sensitive environmental indicators.

### 3.2.1 WAPLS performance; characterising the assemblages

Our first objective is to investigate how the characteristics of the training-set affect the performance of the transfer function.
To achieve the first objective we require robust statistics and so we apply the computationally fast Weighted Average Partial
Least Squares (WAPLS) (ter Braak and Juggins, 1993) and (for simplicity of interpretation) consider only the first PLS
component, equivalent to standard WA with an inverse deshrinking (Birks et al., 1990, 2010). The WAPLS1 data are plotted
as the circles (with associated error bars) in Figure 2. These points are cross-validated (leave-one-out) WA transfer function
performance as a function of the sampling density, defined as the average number of training-set observations per species.
Each plotted circle in Figure 2 is derived from 10,000 training-set sites for each of the three assemblage types. The sites are
randomly generated and assigned to a collection of training-sets of the desired size. For instance, to generate an ensemble of
25-site training-sets, the 10,000 sites are divided into 400 training-sets. This enables us to produce robust statistics for each
data point. Each plotted circle is the mean leave-one-out cross-validated Root Mean Square Error of Prediction (RMSEP) of
the transfer function derived from each ensemble of training-sets, with ±1σ error bars being the standard deviation of the
RMSEP across the ensemble.

Unsurprisingly the low richness-high tolerance training-set is the least well performing. As reconstructions are derived from
an average of only 6 species, they are not well constrained and sensitive to statistical outliers. As the species are typically
characterised by broad tolerance, they are relatively insensitive environmental indicators.  The cross-validated RMSEP
decreases from 16.8±3.3 to 10.0±0.2 as the training-sets increases from 25 to 1000 sites (sampling density increases from 1.4





to 57). The improvement in performance with training-set size is expected as the species optima become more accurately defined. However, there is only a modest improvement in performance as the training-set size increases from 250 to 1000 (sampling density increasing from 14 to 57). These diminishing returns are seen across all three of the assemblage types, and together suggest that a requirement for an average of at least 10 observations per species is a useful indicator for a robustly

characterised training-set, although we note that idealised data are presumably easier to characterise than real (noisier) data. An increase in assemblage richness (by a factor of approximately three) or a decrease in assemblage tolerance (by a factor of two), both approximately halve the WAPLS1 prediction error (RMSEP reductions of between 37% and 52%). We note that while these improvements cannot in general be controlled (they are characteristic of the assemblage), low-richness assemblages are likely to benefit especially from high sample counts (high number of counts per training-set site) which

tends to increase the observed species richness.

### 3.2.2 BUMPER performance and CPU demand

We now consider the performance of BUMPER. The previous applications of BUMPER (to real ecological data) have made

two pragmatic assumptions. First, $\eta = 0.5$ (Equation 5), blending the abundance likelihood function $L_y$ (Equation 3) with the presence/absence likelihood function $L_p$ (Equation 4) so that the wide tails contributed by $L_p$ allow for the possibility of outlying species counts. Second, only including species that have abundances >2% when performing a reconstruction; while very low counts do contribute information, they are generally less informative than high counts, they increase computational load, and they are likely to be less robust. Neither of these conservative assumptions is necessary for application to the

idealized data considered here. However, we choose to retain them for this analysis in order to avoid the risk of overstating the performance statistics (relative to WAPLS) that are likely to be possible in practice.

The crosses on Figure 2 plot the leave-one-out RMSEP of BUMPER. For each of the twelve scenarios we select one training-set, being the first of the randomly generated ensemble members that exhibits a WAPLS1 RMSEP within 0.1 of the

ensemble mean. BUMPER performance is generally comparable to WAPLS1 performance. This has previously been found to be the case in SWAP diatoms (Holden et al., 2008) and tropical-Andean chironomids (Matthews-Bird et al., 2016). We note that individual reconstructions (as distinct from the summary statistics of a training-set) can differ significantly between the two approaches (Matthews-Bird et al., 2016), in particular because of the increased sensitivity of the reconstruction to 'low-count' species (that are only ever observed in low abundance).


The CPU demands of BUMPER are dominated by the calculation of the SRC probabilities (i.e., applying Eq. 1 across all training-set sites for each species). This takes 0.022 seconds per species on a standard personal computer (2.5GHz Intel Core i5 processor, 4 GB 1600 MHz DDR3 memory) for a 100-site training-set. The CPU demand scales approximately linearly with number of sites. To illustrate, deriving the leave-one-out cross-validated performance of a training-set of 500 sites with

an average species richness of 10, would require on average $\sim 4.99 \times 10 \times 0.022 = 1.1$ seconds per site (i.e. to derive leave-one-out SRC probabilities for ten species from 499 training-set sites), thereby taking ~9 minutes to cross-validate the entire training-set.

### 3.2.3 BUMPER uncertainty


The solid lines in Figure 2 plot the BUMPER uncertainty metric Δ (Equation 8) from the reconstructions described in Section 3.2.2. Although the uncertainties broadly reflect the trends in the reconstruction error, uncertainty is significantly overstated, being 151±21% of the RMSEP across the twelve training-sets. This overstatement is expected for idealised data because of





the blended presence/absence likelihood function ($\eta = 0.5$, Equation 5). This intentionally broadens the likelihood functions to prevent an outlier, such as a taxon misidentification, from ruling out the correct solution. While this assumption has proved useful for real data, it is unnecessarily conservative for idealised data and is therefore expected to overstate uncertainty. Additional BUMPER models (not illustrated) were built for each training-set with $\eta = 0.0$ and including all

present species. These models exhibited improved RMSEP relative to the default, reduced by typically 10%. Moreover, these models produced posteriors that better reflect the reconstruction error, with an average uncertainty of 112±17% RMSEP, demonstrating that the uncertainty is meaningfully quantified when the likelihood function is not artificially broadened. (Nevertheless, we retain the broadened likelihood functions for applications to real ecological data.)

**4 Applications to real data**

We consider three training-sets of real data, using three different biological proxies and having different assemblage characteristics, summarized in Table 1. The tropical chironomid (Mean Annual Temperature) dataset of Matthews-Bird et al. (2016) is a 59-site species-poor training-set comprised largely of species with narrow environmental tolerances (expressed as

a percentage of the training-set environmental gradient). The SWAP diatom (pH) dataset (Stevenson et al., 1991) is a 167-site species-rich training-set, significantly larger than the Matthews-Bird dataset, but with broader characteristic tolerances. The NIMBIOS tropical pollen (mean annual temperature) dataset (Bush et al., in prep.) is a 682-site species-rich training-set that is currently under development and characterized by narrow tolerances. These pollen data are not (in their current state) expected to have the high quality of SWAP, because some taxa are identified to family level whereas other taxa are identified

to genus.

We consider four alternative BUMPER models. First, we test the requirement for a 2% abundance threshold (Section 3.2.2) by generating reconstructions with and without this constraint. Removing this constraint is expected to decrease the uncertainty of the reconstruction (all present species are included, thereby narrowing the posterior). We wish to test whether

this decreased uncertainty is associated with a reduction in reconstruction error when applied to real data. Secondly, we consider sensitivity to the form of the likelihood function. The default model applies $\eta = 0.5$ (Section 2.2). However, in Holden et al. (2008), RMSEP was found to be only weakly dependent upon this parameter so that presence/absence data alone ($\eta = 1.0$) contain sufficient information to derive a useful predictive model. Compared with the full model, the presence/absence model is expected to increase robustness at the expense of increased uncertainty, with a broader posterior.

There may be situations where this conservative approach is preferable, for instance when SRCs are poorly constrained in small training-sets, or when misclassification is a concern as the presence/absence model is less sensitive to outliers. The cross-validated performances of the four models ($y_k > 2\%$ or 0%, $\eta = 0.5$ or 1.0) are summarized for each of the three training-sets in Table 2, and are plotted for two models ($\eta = 0.5$ or 1.0, $y_k > 2\%$) in Figure 3.

**4.1 Chironomids (Matthews-Bird et al. 2016)**

The chironomid training-set of Matthews-Bird et al. (2016) comprises 59 lakes across tropical South America. The prediction uncertainty associated with this dataset (WAPLS RMSEP 2.4°C) is approximately twice the uncertainty that has been achieved for European chironomid transfer functions, which typically approach ~1°C (e.g. Brooks et al., 2012). It has been

postulated that this reduced performance is caused by the small size of the training-set and uneven sampling across the environmental gradient (Matthews-Bird et al., 2016).

The default model ($y_k > 2\%$, $\eta = 0.5$) exhibits an RMSEP of 2.41°C. However, the best model (RMSEP 2.27°C) is the presence/absence model that requires 2% abundance to include a species in a reconstruction. This model version is the most





conservative and might therefore be expected to be the least well performing. However, the training-set is small, so the SRCs are not expected to be well defined, and furthermore the number of counts per site is relatively low. These data may therefore favour a more conservative model that does not over-constrain the reconstructions.

When all species are included, the reconstruction uncertainty is reduced as expected, from 12.0% to 11.0% in the default model and from 13.3% to 11.8% in the presence/absence model. However, this is not accompanied by a reduction in RMSEP. The RMSEP increases from 9.7% to 9.8% in the default model and from 9.1% to 9.6% in the presence/absence model. These data support our choice to retain the 2% abundance threshold in the default model. Including very low abundance counts increases the computational demand, does not in general improve model performance, and is likely to

under-state the uncertainty associated with the reconstruction. We note for completeness that the default BUMPER model was also applied in Matthews-Bird et al. (2016) and has a RMSEP of 2.37°C. The slight difference in performance here results from the different SRCs priors used (see Section 2.4).

    The Matthews-Bird training-set can be broadly classified as being low species-richness (7.0), narrow tolerance (12%), and

relatively poorly sampled (sampling density 7.0). We generated an ensemble of artificial datasets with approximately those characteristics (59 lakes, 59 taxa, $\beta_p = 40\%$, $t_L = 8\%$, $t_H = 18\%$, yielding an ensemble averaged species richness of 6.7). The WAPLS1 transfer functions built from these data exhibits a mean RMSEP of 6.3±0.6% of the environmental gradient. Although a direct comparison is difficult, given that an improvement in performance is expected under idealized assumptions, these statistics are broadly consistent with those of the real transfer functions (ranging from 9.1% to 9.8%).

    We consider the training-set characteristics in order to investigate whether they might explain the larger uncertainties of the Matthews-Bird transfer function when compared with European chironomid transfer functions. First, we increase the idealized-analogue training-set size from 59 lakes to 118 lakes, thereby better defining the species characteristics. This improves the WAPLS1 RMSEP from 6.3±0.6% to 5.9±0.4%. Second, we increase the species richness from 6.7 to 13.5 by

doubling the number of species to 118. This improves the WAPLS1 performance to 5.1±0.5%. Although both factors are likely to contribute to explaining the reduced uncertainty of the European transfer functions, in isolation they do not appear sufficient. We consider a specific example. The Norwegian-chironomid training-set of Brooks et al. (2012) comprises 140 taxa sampled from 157 lakes, and was used to construct a WAPLS2 transfer function for July temperature with RMSEP of 1.06°C when four outliers were deleted (Brooks et al., 2012). We note that the Matthews-Bird training-set reconstructs MAT,

not July temperature. The Brooks and Birks training-set has an average richness of 24 and an indicative tolerance of 16%. An idealized dataset was built with these characteristics (159 lakes, 140 taxa, $\beta_p = 55\%$, $t_L = 11\%$, $t_H = 21\%$, yielding an ensemble averaged species richness of 22) and was found to exhibit an WAPLS1 RMSEP of 5.1±0.3% (=0.64°C). When expressed as a percentage of the environmental gradient, modest improvements in performance are apparent, consistent with that expected from increased training -set size and species richness. However, most of the improvement in predictive power

is due to tolerance, but expressed in *absolute* terms. The Matthews-Bird indicative tolerance is 3.2°C. The Brooks and Birks indicative tolerance is 2.1°C. These factors together (increased training-set size, increased species richness and, most significantly, narrower *absolute* tolerances) appear sufficient to explain the improved performances of the Brooks and Birks transfer function over the Matthews-Bird transfer function.

**4.2 SWAP (Stevenson et al. 1991)**

    The SWAP training-set (Stevenson et al., 1991) was developed as part of a substantial scientific effort directed at understanding the impacts of acid rain. Taxonomic workshops resolved problems of nomenclature, splitting and amalgamation of species, and identification criteria (Munro et al., 1990). Approximately 500 counts per sample were made.





The training-set was statistically pruned (Birks et al., 1990) to leave a high-quality dataset of 267 taxa in 167 sites. The best WAPLS component has RMSEP of 0.310 pH units (Holden et al., 2008).

The best-performing BUMPER model for SWAP is the model that includes all species, with no threshold abundance. This model has an RMSEP = 0.321 pH units. The improvement in performance relative to the default BUMPER ($y_k > 2\%$) threshold (RMSEP = 0.369pH units) may reflect the high quality of the SWAP training-set, so that even very low counts are unlikely to be erroneous, and therefore add value to the reconstruction. However, a weakness of the $y_k > 0\%$ model is that the posterior widths substantially understate the uncertainty; the low-count taxa narrow the posterior widths by more than is justified, given the modest improvement in performance. We note that the performance of the subjectively tuned model in

Holden et al. (2008) was RMSEP = 0.328 pH units.

The SWAP training-set can be broadly classified as being high species-richness (50), broad tolerance (22%) and well sampled (sampling density 37). It is interesting that the RMSEP of this training-set is similar to the Matthews-Bird set, both exhibiting an RMSEP of approximately 10% of the environmental gradient. The improvements due to the high quality,

species richness, and sampling density of SWAP are offset by the loss of precision that is achievable due to broader tolerances.

We generate an artificial training-set to mimic the characteristics of SWAP (167 lakes, 225 taxa, $\beta_p = 40\%, t_L = 17\%,$ $t_H = 27\%$, yielding a characteristic species richness of 48). The performance statistics of a model derived from these data

5.1±0.3% are again broadly consistent with, though significantly better than, the model derived from the real data (11.0 to 12.9%). The difference between real and artificial data is greater than in the chironomid comparison. Diatoms have been applied as indicators of a wide range of environmental variables including pH, climate, water chemistry, and eutrophication (Smol and Stoermer, 2015), and it may be that the assumption of a single environmental driver of diatom assemblage variability is less well satisfied than it is with chironomids.


### 4.3 NIMBIOS

The NIMBIOS dataset (Bush et al., in prep) is comprised of 682 samples that range from soil samples to mud-water interface samples from lakes. The dataset is of variable quality as it represents a ~30–year data acquisition effort in which time there

have been substantial improvements in the ability to identify accurately pollen because new pollen keys and descriptions have become available (e.g. Roubik and Moreno, 1991; Bush and Weng, 2007).

The best BUMPER model for NIMBIOS is the full model including all taxa ($\eta = 0.5, y_k > 0\%$), exhibiting an RMSEP = 2.51°C. This represents a significant improvement to the best WAPLS model (2.92°C, WAPLS component 2). As with

SWAP, however, a weakness of this model is that the posteriors substantially understate the uncertainty. The default model also performs well (RMSEP = 2.88°C) and exhibits posteriors that accurately reflect the uncertainty.

The NIMBIOS training-set can be classified as being high richness (27), low tolerance (12%) and well sampled (sampling density 33). The training-set benefits from optimal characteristics in all three respects. We generate an artificial analogue of

NIMBIOS (682 samples, 553 taxa, $\beta_p = 15\%, t_L = 7\%, t_H = 17\%$, yielding a species richness of 30). The BUMPER model built from this analogue exhibits an RMSEP of 3.8±0.1%. This is significantly better than the artificial analogues of the SWAP and Matthews-Bird sets, as would be expected given the optimal characteristics of the training-set. However, the performance statistics in this case are substantially better than those of the model built from the real data (RMSEP = 10.1%




to 12.2%); the transfer function built from real data does not demonstrate the high performance that might be expected. The discrepancy between the idealised and real data appears to suggest that the quality of the NIMBIOS data is poorer than the other training-sets. A major reason for the apparent inaccuracy may be that, unlike the other datasets, pollen is often not recognised to species level[3] and so a pollen type could contain numerous species that have different environment preferences, and hence may well have a multimodal distribution rather than the unimodal distribution assumed by BUMPER.

**5 Summary and Conclusions**

We have developed BUMPER, a user-friendly implementation of a Bayesian transfer function for palaeoenvironmental reconstruction. BUMPER is fully self-tuning, applying a precalibration approach to eliminate implausible values for five model parameters before performing the full probabilistic calibration. The model is straightforward to use, and the only requirement for application to a new training-set is to construct the input data-files (as described in Appendix 1).

We have applied BUMPER to generate transfer functions for chironomid-based temperature (Matthews-Bird et al., 2016), diatom-based pH (Stevenson et al., 1991), and pollen-based temperature (Bush et al., in prep). In each case cross-validated performance statistics are comparable to the best WAPLS component. The motivation for the model is that it generates a reconstruction-specific uncertainty that, for instance, renders it well suited to multi-proxy reconstructions.

Although the model calibration is performed using all observations, including species absence, for reconstruction purposes we only consider species found at abundances greater than 2%. When species with very low counts are included in a reconstruction, the additional narrowing of the posterior is found to exceed the reduction in reconstruction error (with the Matthews-Bird training-set it even leads to increased error), suggesting that the information provided by very low counts is less reliable. This conclusion is supported by application to idealized artificial data when the reconstruction uncertainty is found to reflect accurately the reconstruction error when all species are included (and when the likelihood function is not artificially broadened). We retain the 2% threshold for application to real data as we regard the reliable quantification of uncertainty to be more important than the modest improvements in prediction accuracy than can be achieved from including very low species counts.

The performance statistics of the three datasets are quite similar, with each displaying a cross-validated error of prediction that is approximately 10% of the environmental gradient. However artificial training-sets with similar characteristics to the real data revealed that this similarity is largely coincidental and arises for different reasons. The chironomid data are hampered by species-poor assemblages, but benefit from narrow tolerance species. Conversely, the diatom data are composed of species with broader tolerances (expressed as percentage of environmental gradient) but the assemblages are substantially richer and species are well characterized by the large dataset. The pollen data are large, taxon-rich, and characterized by narrow tolerances. The pollen data were expected to outperform significantly the other training-sets, however, this is not found to be the case. The likely explanation is that pollen, unlike the other organism-types considered, can be difficult to identify to species level.

**Code Availability**

The source code BUMPER1p0.F90, example input data files (Matthew-Bird et al, 2016) and an output visualizer BUMPERpp.R are all provided as supplementary material. Instructions are provided in the Appendices.

---

[3] We note that chironomids are identified to species morphotype level, or genus in some cases.


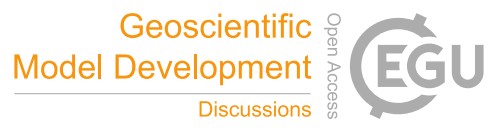

**Acknowledgements**

PBH was funded through the EPSRC project "Research on Changes of Variability and Environmental Risk" (ReCoVER),
award RFFER003. MBB, GMH, BGV and RvW were funded through the Climate Proxies Working Group at the National
Institute for Mathematical and Biological Synthesis, sponsored by the National Science Foundation through NSF Award
#DBI-1300426, with additional support from The University of Tennessee, Knoxville. FMB was funded by the National
Aeronautics and Space Administration (NASA) NNX14AD31G.

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

**Appendix 1: USER MANUAL**

The authors are happy to provide whatever assistance may be needed to run BUMPER. The model is written in FORTRAN
90, included as supplementary material BUMPER1P0.F90.  The open source GNU Fortran compiler is freely available for MacOS, Linux, and Windows platforms. Once a compiler has been installed, the model is compiled to create an executable BUMPER with the (linux) command

```
$ gfortran BUMPER1P0.f90 –o BUMPER
```

Before running BUMPER, four input data files must be produced and saved in the same directory as the executable file. These files are NAME.lakes, NAME.species, NAME.ts.counts and NAME.core.counts, where "NAME" should be set to a string that identifies the dataset. The Matthews-Bird data have been uploaded to supplementary information as an example "MB16".


MB16.lakes: the first line of this file is the number of training-set lakes and is followed by a list of site names together with the environmental variable. All string inputs are limited to 20 characters. Do not include spaces (use an underscore) in character strings. Avoid characters with functionality in R, such as "/", as these will upset the BUMPERpp plotting software (Appendix 2).



MB16.species: the first line is the number of species in the training-set, and this is followed by a list of species names.

MB16.ts.counts: a file providing the percentage counts of all species in all training-set sites. The file consists of a row for
each lake, each row containing a list of the species counts. It is very important that the rows are ordered identically as they
are listed in NAME.lakes and the columns are ordered identically as they are in NAME.species.

MB16.core.counts has the same format as MB16.ts.counts, except the first line is the number of core samples and the first
column of the subsequent data is the name of the sample (e.g. depth or age).

The model, can now be run, using the command

$ ./BUMPER

The first things that the model will ask for is the name of the training-set, in our example MB16. You will then be asked a
series of questions:

     1)   whether to calculate the apparent RMSE (i.e. the fitted model error)

     2)   whether to calculate the jackknifed (leave-one-out) RMSEP

     3)   whether to reconstruct the core

4)   whether to build a presence/absence model (0==NO=preferred default)

     5)   whether to apply the 2% abundance threshold (1==YES=preferred default)

During the input phase, the model will provide the sampling density, mean richness, and indicative tolerance as outputs to the
screen. After the input phase, the requested calculations will be performed and various summary outputs will be provided to
the screen as the calculations progress. Output files will be created, depending upon the options chosen. These comprise

1) output.data: summary performance statistics, together with individual site reconstructed values and uncertainties

2) ts_likelihood_app.data: likelihood functions for all species and all training-set sites from the fitted model.

3) ts_likelihood_core.data: jackknifed likelihood functions for all species and all training-set sites (i.e. cross-validated
likelihood functions)

4) core_likelihood.data: likelihood functions for all species and all core samples

5) reconstruct_ts_app.data: the posteriors of the fitted model

6) reconstruct_ts_jack.data: the (cross-validated) posteriors of the jack-knifed models

7) reconstruct_core.data: the core sample posteriors

**Appendix 2: Data visualization with BUMPERpp**

The self-documented tool BUMPERpp.R is a post-processing graphical tool written in R, included as supplementary
material. The code reads the likelihood data files and generates plots of the reconstructions.

Load the model in R with the command

> source("BUMPERpp.R")

Generate a plot with the (illustrative) command

> BUMPERpp("ts_likelihood_jack.data",1)



i.e. specifying the input file and sample number to be plotted.

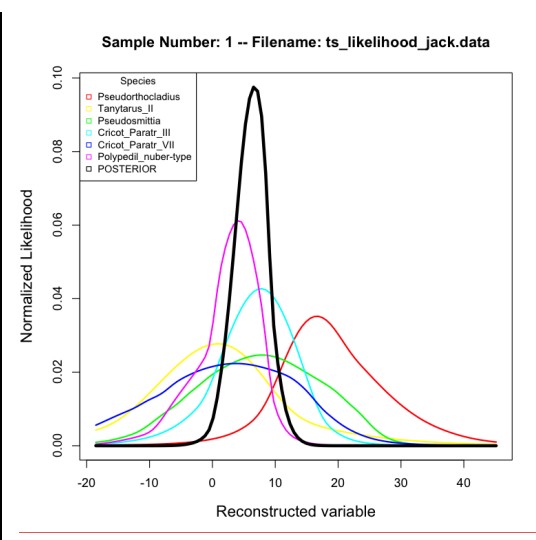



**TABLE 1: Training-set characteristics**

|  | CHIRONOMIDS | DIATOMS | POLLEN |
|---|---|---|---|
| **Env. Gradient** | 25.0°C | 2.92pH units | 24.9°C |
| **Number of sites** | 59 | 167 | 682 |
| **Number of species** | 59 | 225 | 553 |
| **Average Richness** | 7.0 | 50 | 27 |
| **Indicative tolerance** | 3.2°C /13% | 0.63 / 22% | 3.0°C /12% |
| **Sampling density** | 7.0 | 37 | 33 |

5 **TABLE 2: Four BUMPER models.** For each of the three training-sets (Matthews-Bird et al., 2016, Stevenson et al., 1991, Bush in prep.) four models are considered. We use either the standard model ($\eta = 0.5$) or the presence absence model ($\eta = 1.0$) and reconstruct with either all species ($y_k > 0\%$), or only including species at more than 2% abundance ($y_k > 2\%$).

| 1) $y_k > 0\%, \eta = 0.5$ | CHIRONOMIDS | DIATOMS | POLLEN |
|---|---|---|---|
| **RMSEP** | 2.46°C / 9.8% | 0.321 pH / 11.0% | 2.51°C / 10.1% |
| **POSTERIOR WIDTH** | 2.74°C / 11.0% | 0.183 pH /6.2% | 1.64°C / 6.6% |

| 2) $y_k > 0\%, \eta = 1.0$ | CHIRONOMIDS | DIATOMS | POLLEN |
|---|---|---|---|
| **RMSEP** | 2.40°C / 9.6% | 0.357 pH / 12.2% | 2.73°C / 11.0% |
| **POSTERIOR WIDTH** | 2.95°C / 11.8% | 0.204 pH / 7.0% | 1.73°C / 6.9% |

| 3) $y_k > 2\%, \eta = 0.5$ | CHIRONOMIDS | DIATOMS | POLLEN |
|---|---|---|---|
| **RMSEP** | 2.41°C / 9.7% | 0.369 pH / 12.6% | 2.88°C / 11.6% |
| **POSTERIOR WIDTH** | 3.01°C / 12.0% | 0.330 pH / 11.3% | 2.79°C / 11.2% |

| 4) $y_k > 2\%, \eta = 1.0$ | CHIRONOMIDS | DIATOMS | POLLEN |
|---|---|---|---|
| **RMSEP** | 2.27°C / 9.1% | 0.377 pH / 12.9% | 3.04°C / 12.2% |
| **POSTERIOR WIDTH** | 3.33°C / 13.3% | 0.473 pH / 16.1% | 3.77°C / 15.1% |



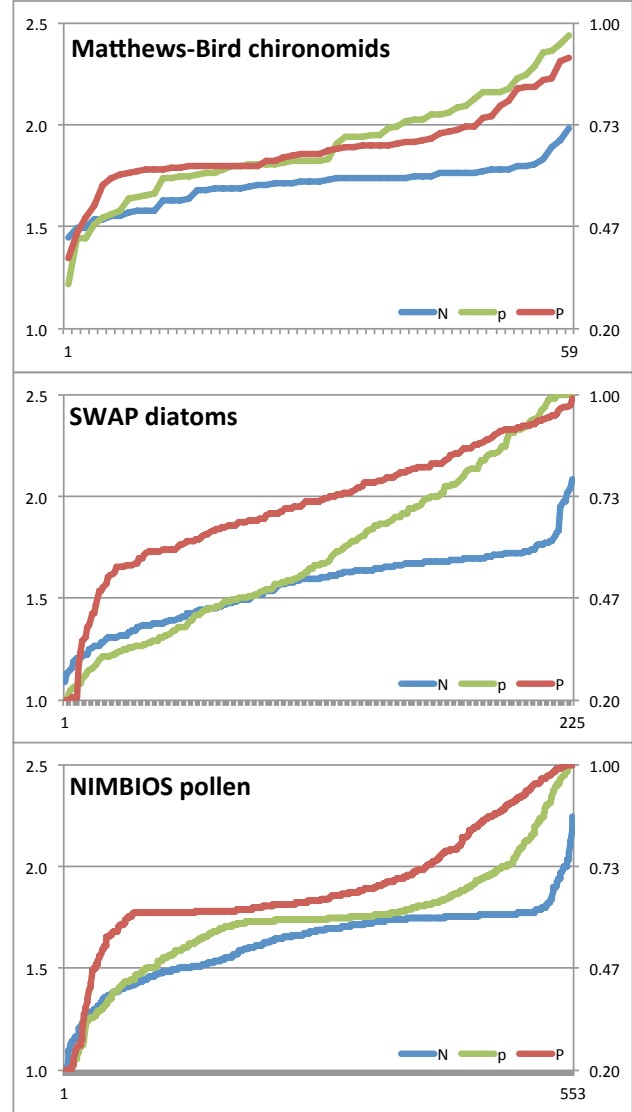

**Figure 1: Probability-weighted values of the SRC parameters $N_k$ (left-hand axis), $p_k$ (left-hand axis) and $P_k$ (right-hand axis). Three training-sets are considered: chironomid-based temperature (Matthews-Bird et al., 2016), diatom-based pH (Stevenson et al., 1991) and pollen-based temperature (Bush et al., in prep). For each SRC parameter, probability-weighted values across for the taxa of each training-set are ordered and plotted sequentially. Horizontal gridlines represent the discrete values allowed within individual SRCs.**





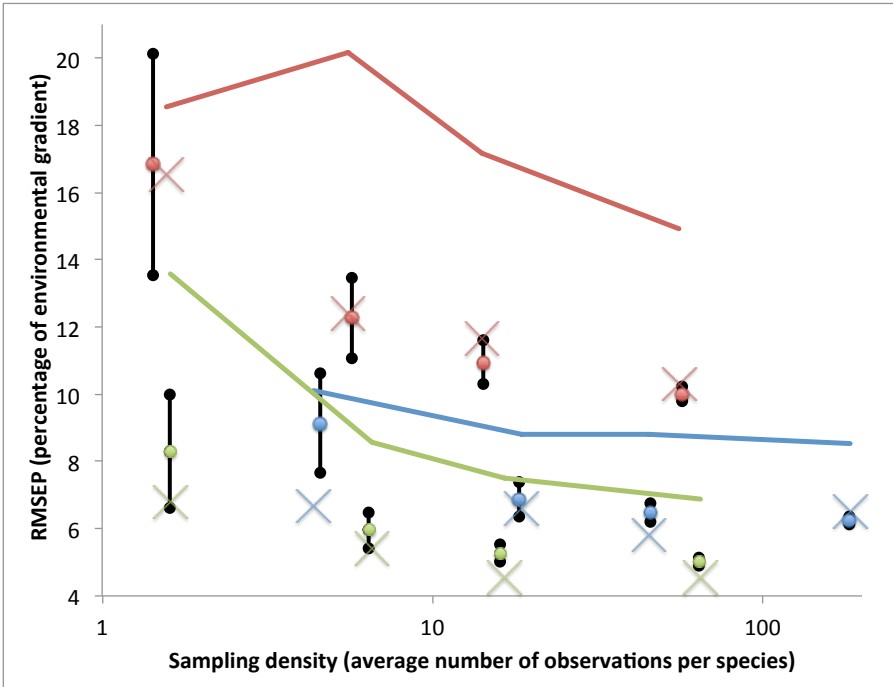

Figure 2: Reconstruction performance plotted against the sampling density (the average number of observations per species). Colour indicates assemblage type, being low richness, high tolerance (red), high richness, high tolerance (blue) or low richness, low tolerance (green). Circles with error bars are the mean and standard deviation of the WAPLS1 leave-one-out RMSEP calculated across an ensemble of artificial training-sets. From each artificial ensemble, a member is selected that has a WAPLS1 error equal to the ensemble mean. BUMPER is applied to this training-set; crosses are the BUMPER leave-one-out RMSEP, solid lines are the BUMPER uncertainty.



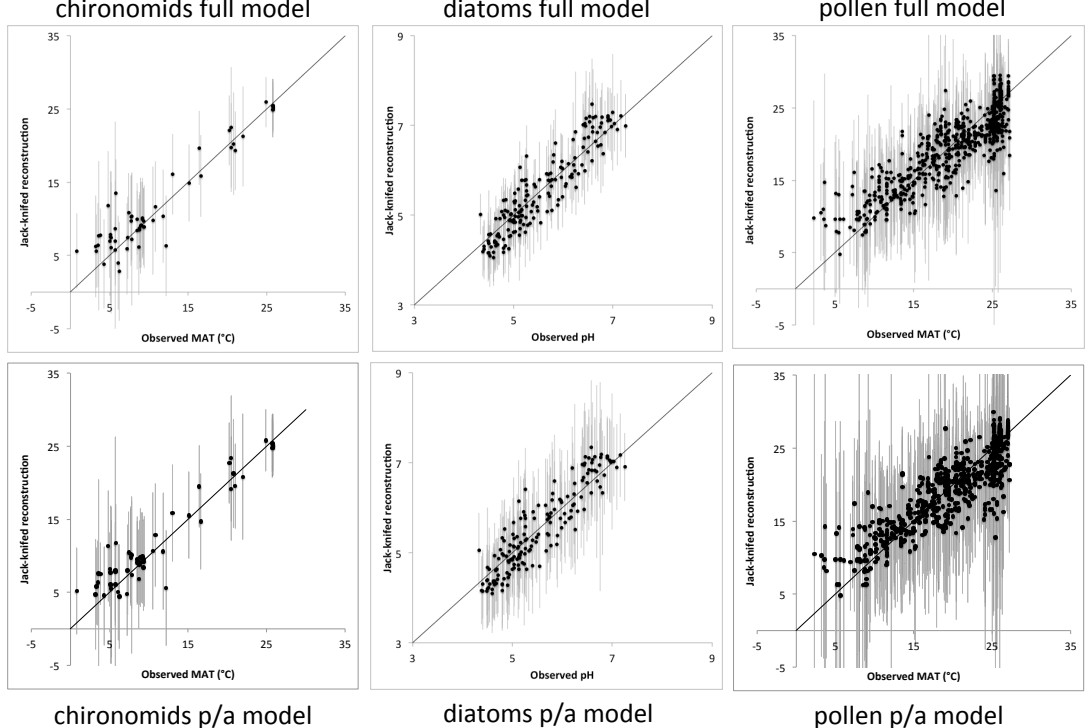

**Figure 3. Cross-validated reconstructions (Eq. 7) together with associated uncertainty (Eq. 8) are plotted against observed training-set environment for each of the three training-sets (Matthews-Bird et al., 2016; Stevenson et al., 1991; Bush et al., in prep.) We plot the full model ($\eta = 0.5$) and the presence/absence model (p/a) ($\eta = 1.0$). All analyses impose the constraint that 2% abundance is required for inclusion.**