# Peer review of "BUMPER v1.0: A Bayesian User-friendly Model for Palaeo-Environmental Reconstruction"

_Geoscientific Model Development, 2016_

## Referee Comment (RC1) · C. ter Braak (Referee) · 31 Oct 2016

General comments

The first aim of this paper is to alleviate a difficult problem in every Bayesian analysis: that of setting priors. In this case for the Bayesian approach to palaeo-environment reconstruction using Gaussian response curves. This approach was first proposed in Holden et al. 2008. The priors are partly based on the data to be analyzed (as in Empirical Bayes), so that the approach is not fully Bayesian in the strict sense; but it practical and appealing.

The second aim is to provide a more general evaluation of this approach, with the new

priors, to simulated data sets and a number of (famous and newer) data sets. The approach is compared with the simple approach based on weighted averaging with deshrinking using inverse regression, aka WAPLS1.

The computation approach avoids the usual MCMC computation, or approximations thereof (e.g. INLA), in Bayesian analysis by limiting the approach to one-dimensional modelling and reconstruction and by discretising the parameters, so that in fact 2,560 possible parameter combinations remain. Thereby a fully Bayesian analysis is possible without MCMC. The posteriors for these models act as if they are weights in a model averaging exercise. It is well known that simple models when averaged can solve complex problems.

In my view, the paper fits in the journal, has a clear aim and fulfils the claims.

Specific comments

In one place, it looks like the distinction between prior and posterior is lost in the notation/formulas. In section 2.2 prob(SRC_jk) is surely the posterior denoted by prob(SRC_jk|Y,X), where Y and X are the training data (as in eq (1)). Note that the model setup also belongs to the condition. In eq( 2), the posterior weights are meant, is it not? Instead of adapting all formulas (when I am right in this) state explicitly that "From now on prob(SRC_jk) is the posterior probability, the probability of the SRC given the training data Y and X.

The probability distributions in section 2.3 form a hurdle model (zero inflated distribution wit truncation at 0 of the count distribution. If I am right in this, please mention this.

I numbered the pages from 1-20.

P2L4: in the model? It depends of course what you mean with model here. But in the natural sense, the model is fixed and only the parameters of the model are uncertain, and for discrete parameters their distribution (weights). So, say so. Even although the approach has aspects of model averaging, it is best viewed as defining one model...,

which is then fixed.

P4L14 (end section 2.2) At how many points is x evaluated?

P4L19 The expected abundance follows a distribution??? This is not a distribution in any of the senses you use in this paper; it is a Gaussian response curve model; see ter Braak & Barendregt 1986 http://dx.doi.org/10.1016/0025-5564(86)90031-3 when it has aspects of a distribution.

P4L39-41. Eq (11)Note it relation to the exponential distribution and geometric distribution. https://en.wikipedia.org/wiki/Exponential_distribution. Probably you treat is as a discrete distribution and truncate is at 0 (or y<1).

On P5L1 we learn that you made an assumption on the data y: between 0 and 100. Please be more explicit and/or give a more general denominator in (11). Such things can lead to strange errors later on, when used without further scrutiny.

P13L40 To make the program even more user-friendly a wrapper in R and/or Python appears much wanted. Make it a priority.

Technical corrections

P2L21. The two -> Two (they were not mentioned before)

Cajo ter Braak, Wageningen , 31 October, 2016

―――――――――――――――――

---

## Referee Comment (RC2) · A. Parnell (Referee) · 16 Nov 2016

This paper introduces a Bayesian model (nicely acronymed 'BUMPER') to reconstruct palaeo-environmental variables from various proxies given modern training data and fossil samples. The model contains two neat ideas which I haven't seen before, namely that of using a mixture likelihood to model both abundance and presence/absence, and the idea of scoring training sets according to their richness and diversity.

I do have concerns about the mathematical model and the way it is described. As somebody who lives and breathes these types of models I found the mathematics confusing and my guess is that they will go straight over the head of the average reader of this journal.

- Starting with Equation 1, it seems to be missing a product term, which I think should appear after the proportionality constant. Either that or each SRC is being calculated for each taxa, species and site combination. This seems unlikely.

- Equation 2 seems to suggest that this is the normalising constant, but that can't be the case as the Bayes equation is $p(SRC|y) = p(y|SRC)p(SRC)/p(y)$. It's $p(y)$ that needs to be in the normalising constant.

- Equations 3 and 4 suggest that the likelihoods are all only known up to proportionality and the proportionality component isn't mentioned. I think these should all be equals signs.

- Equation 6 suggests that there is another Bayesian model being fitted. It's thus not clear whether there is one model being fitted (which is all that is required) or whether multiple Bayesian models are being stitched together.

All this points to a more fundamental problem, namely that of the lack of a statistical collaborator. These authors are world-renowned experts in the field of collecting and understanding the nuances of proxy data and how it links with climate. There are statisticians and groups out there (for example the Past Earth Network) who can help.

Models like these are now being studied by statisticians in collaboration with proxy experts. One that is not yet in the palaeoclimate literature (which is perhaps why the authors might have missed it) is that of Ilvonen et al which seems very similar to what the authors are trying to achieve here. A more flexible version can be found in Cahill et al which is in the palaeoclimate literature and uses multiple proxies (forams and d13C) in a Bayesian model for sea level reconstruction. A more recent model is my own Bclim (Parnell et al) which allows for joint inference (i.e. all fossil slices, all taxa, multiple climate variables) to be estimated together, with the aim of reducing uncertainty.

Lastly a note on the figures. Again I found these hard to follow. Figure 1 has three lines on three panels with two different y-axes. The x-axis runs from different values for

each of the three panels. My guess is that this is each SRC above the 1% threshold but it's not clear. I found matching this to the text very hard. Figure 2 contains lines, candlesticks, points, crosses, and three different colours. I've read the caption multiple times, and the text associated with it (which covers 3 different sections), and still cannot work out what's being learnt from this picture. Figure 3 is much more useful, but seems to be hardly mentioned aside from the end of the last paragraph of Section 4. It was a shame not to see any actual reconstructions of climate over time for any of the sites.

References used:

Ilvonen, L., Holmström, L., Seppä, H., and Veski, S. (2016) A Bayesian multinomial regression model for palaeoclimate reconstruction with time uncertainty. Environmetrics, 27: 409–422. doi: 10.1002/env.2393.

Cahill, N., Kemp, A. C., Horton, B. P., & Parnell, A. C. (2016). A Bayesian hierarchical model for reconstructing relative sea level: from raw data to rates of change. Climate of the Past, 12(2), 525-542.

Parnell, A. C., Haslett, J., Sweeney, J., Doan, T. K., Allen, J. R., & Huntley, B. (2016). Joint palaeoclimate reconstruction from pollen data via forward models and climate histories. Quaternary Science Reviews, 151, 111-126.

---

## Author Comment (AC1) · 7 Dec 2016

**BUMPER, Holden et al, GMDD**
**Response to C. ter Braak (Referee)**

**Referee comments black.**
**Author responses red.**
**Manuscript changes green.**

We are grateful for this careful review, which has certainly improved the mathematical rigour of the model description.

General comments

The first aim of this paper is to alleviate a difficult problem in every Bayesian analysis: that of setting priors. In this case for the Bayesian approach to palaeo environment reconstruction using Gaussian response curves. This approach was first proposed in Holden et al. 2008. The priors are partly based on the data to be analyzed (as in Empirical Bayes), so that the approach is not fully Bayesian in the strict sense; but it practical and appealing.

The second aim is to provide a more general evaluation of this approach, with the new priors, to simulated data sets and a number of (famous and newer) data sets. The approach is compared with the simple approach based on weighted averaging with deshrinking using inverse regression, aka WAPLS1.

The computation approach avoids the usual MCMC computation, or approximations thereof (e.g. INLA), in Bayesian analysis by limiting the approach to one-dimensional modelling and reconstruction and by discretising the parameters, so that in fact 2,560 possible parameter combinations remain. Thereby a fully Bayesian analysis is possible without MCMC. The posteriors for these models act as if they are weights in a model averaging exercise. It is well known that simple models when averaged can solve complex problems. In my view, the paper fits in the journal, has a clear aim and fulfils the claims.

Specific comments

In one place, it looks like the distinction between prior and posterior is lost in the notation/formulas. In section 2.2 prob(SRC_jk) is surely the posterior denoted by prob(SRC_jk|Y,X), where Y and X are the training data (as in eq (1)). Note that the model setup also belongs to the condition. In eq( 2), the posterior weights are meant, is it not? Instead of adapting all formulas (when I am right in this) state explicitly that "From now on prob(SRC_jk) is the posterior probability, the probability of the SRC given the training data Y and X.

Yes, we agree and have made this change (correcting Eq. 2, and after then adding the suggested sentence). The mathematics should also be simpler to follow now we have changed Eq. 1 to reflect the application of the entire training set (see response to the other referee, Andrew Parnell) . Eq 1 and 2 now read

$$prob(SRC_{jk}|Y_k, X) \propto prob(SRC_{jk}) \times \prod_i prob(y_{ik}|SRC_{jk}, x_i) \qquad \text{Eq. 1}$$

$$\sum_j prob\left(SRC_{jk}|Y_k, X\right) = 1 \qquad\qquad\qquad\qquad \text{Eq. 2}$$

The probability distributions in section 2.3 form a hurdle model (zero inflated distribution wit truncation at 0 of the count distribution. If I am right in this, please mention this.

Thank you, we have made this clarification.

"Species counts distributions are represented with a hurdle model (a zero inflated distribution with truncation at zero of the distribution of percentage counts)."

I numbered the pages from 1-20.

P2L4: in the model? It depends of course what you mean with model here. But in the natural sense, the model is fixed and only the parameters of the model are uncertain, and for discrete parameters their distribution (weights). So, say so. Even although the approach has aspects of model averaging, it is best viewed as defining one model..., which is then fixed.

Agreed. We have changed the text ("uncertainty in the data and in the model parameters") to clarify that we are referring only to the parametric uncertainty of the model here (not the structural uncertainty arising from choice of mathematical structure). Additionally, we have changed the description of the parameter selection process from "model selection" to "parameter estimation".

P4L14 (end section 2.2) At how many points is x evaluated?

New text added

"The likelihood functions are evaluated at 100 evenly spaced points across a range that comfortably spans the training set environmental range (see section 2.4), and are normalised to 1."

and

"The reconstruction is evaluated at the same 100 points as the likelihood function, and the probabilities normalised to 1."

and (in Section 2.4)

"We note that the indicative tolerance is also used to define the range of environment considered in the reconstruction (Section 2.2), from $(x_{min} - 6t')$ to $(x_{max} + 6t')$. Significant probabilities beyond this range are unlikely given the constraints imposed upon the optima and tolerances. In any event, as

with any transfer function, the model should not be applied under suspected extrapolation beyond the training set environment."

P4L19 The expected abundance follows a distribution??? This is not a distribution in any of the senses you use in this paper; it is a Gaussian response curve model; see ter Braak & Barendregt 1986 http://dx.doi.org/10.1016/0025 5564(86)90031-3 when it has aspects of a distribution.

Change made and citation added

"can be fitted by a Gaussian response curve (ter Braak and Barendregt, 1986)"

P4L39-41. Eq (11) Note it relation to the exponential distribution and geometric distribution. https://en.wikipedia.org/wiki/Exponential_distribution. Probably you treat is as a discrete distribution and truncate is at 0 (or y<1). On P5L1 we learn that you made an assumption on the data y: between 0 and 100. Please be more explicit and/or give a more general denominator in (11). Such things can lead to strange errors later on, when used without further scrutiny.

Clarified

"is expressed as a continuous distribution, truncated at 0 and 100:"

P13L40 To make the program even more user-friendly a wrapper in R and/or Python appears much wanted. Make it a priority.

Thank you for this suggestion, we will look into this.

Technical corrections

P2L21. The two -> Two (they were not mentioned before)

Change made

---

## Author Comment (AC2) · 7 Dec 2016

**BUMPER, Holden et al, GMDD**
**Response to A. Parnell (Referee)**

**Referee comments in black.**
**Author responses in red.**
**Manuscript changes in green.**

We are grateful for this careful review, which has certainly improved the clarity of both the model description and presentation of results.

This paper introduces a Bayesian model (nicely acronymed 'BUMPER') to reconstruct palaeo-environmental variables from various proxies given modern training data and fossil samples. The model contains two neat ideas which I haven't seen before, namely that of using a mixture likelihood to model both abundance and presence/absence, and the idea of scoring training sets according to their richness and diversity.

I do have concerns about the mathematical model and the way it is described. As somebody who lives and breathes these types of models I found the mathematics confusing and my guess is that they will go straight over the head of the average reader of this journal.

We believe the confusion was most likely due to misleading notation in Eq 1 and Eq 2, and we are grateful to the referee for pointing this out. We have corrected this as detailed below.

- Starting with Equation 1, it seems to be missing a product term, which I think should appear after the proportionality constant. Either that or each SRC is being calculated for each taxa, species and site combination. This seems unlikely.

Our approach in Eq 1 was an attempt to describe how the SRC probabilities are progressively refined by sequential consideration of each training site. i.e. we were describing the calculation of prob(SRC_jk| y_ik, x_i), not prob(SRC_jk| Y_k, X). The accompanying text used to read "*Equation 1 is applied sequentially across all training-set sites. At each application, the posterior derived from the previous training-set site becomes the prior for application to the next site, so that the probabilities assigned to each SRC become progressively better defined.*"

However, we agree this has resulted in imprecise and confusing notation. As suggested by the referee, we have revised Eq 1 to express the product explicitly

$$prob(SRC_{jk}|Y_k, X) \propto prob(SRC_{jk}) \times \prod_i prob(y_{ik}|SRC_{jk}, x_i)$$

and have modified the text accordingly (the descriptive text in italics above is no longer required).

- Equation 2 seems to suggest that this is the normalising constant, but that can't be the case as the Bayes equation is p(SRC|y) = p(y|SRC)p(SRC)/p(y). It's p(y) that needs to be in the normalising constant.

Apologies, this is a consequence of the same confusing notation, i.e. using prob(SRC_jk) interchangeably for the prior and the posterior. We have corrected this to show that it is the posteriors that are normalised using the constraint:

$$\sum_j prob(SRC_{jk}|Y_k, X) = 1$$

After this point we revert to the prob(SRC_jk) notation, with clarification as suggested by the other referee, Cajo ter Braak. "From now on $prob(SRC_{jk})$ is the posterior probability, the probability of the SRC given the training data Y and X."

- Equations 3 and 4 suggest that the likelihoods are all only known up to proportionality and the proportionality component isn't mentioned. I think these should all be equals signs.

We have clarified this with additional text. We prefer to use normalised likelihood functions (and now name them accordingly) so that we can regard them as pdfs. In effect we are describing the reconstruction that would be derived from a single species with a uniform prior.

To reconstruct the environment from an observed fossil count $y_{k0}$ of each taxon $k$ within the fossil assemblage, the probability-weighted SRCs are used to derive likelihood functions for each taxon of the reconstructed variable $x$. Here we again use the Bayes relationship, this time stating that the probability that a reconstructed value is correct in the light of an observed species count is proportional to the probability that the species count would be observed in that environment. Considering a single observed species in isolation, Bayes' equation can be written:

$$prob(x|y_{k0}) \propto prob(y_{k0}|x) \times prob(x) \tag{3}$$

We derive a normalised likelihood function for the taxon, considering only the first term on the right hand side of Eq. 3. This allows us to treat the function as a probability distribution of the environmental variable, given no prior knowledge and a count of this single taxon in isolation. As we do not know the true SRC of the taxon with certainty (in general, the calibration will have resulted in many SRCs with non-zero probability), the likelihood function is derived from all significant SRCs, combined using the probability weights calculated in Section 2.1, as follows:

$$L_y(x|y_{k0}) \propto prob(y_{k0}|x) = \sum_j\{prob(SRC_{jk}) \times prob(y_{k0}|SRC_{jk}, x)\} \tag{4}$$

This expression is evaluated at 100 evenly spaced points across a range that comfortably spans the training set environmental range (see section 2.4), and is normalised to 1. It defines the probability density function of the environment, given no prior information and some observed count $y_{k0}$ of species $k$.

And, at the end of section 2.4

"We note that the indicative tolerance is also used to define the range of environment considered in the reconstruction (Section 2.2), from $(x_{min} - 6t')$ $to$ $(x_{max} + 6t')$. Significant probabilities beyond this range are unlikely given the constraints imposed upon the optima and tolerances. In any event, as with any transfer function, the model should not be applied under suspected extrapolation far beyond the training set environment."

 - Equation 6 suggests that there is another Bayesian model being fitted. It's thus not clear whether there is one model being fitted (which is all that is required) or whether multiple Bayesian models are being stitched together.

Separate models are fitted for all species k (Eq 1). Equation 6 (now Eq 7) describes how the likelihood functions generated from all considered species are combined to with a prior for the environment to generate the reconstruction. Additional text to clarify

"The posterior probability distribution for the reconstructed variable is derived by combining any prior knowledge with the product of likelihood functions of all considered species in the assemblage, as follows:"

 All this points to a more fundamental problem, namely that of the lack of a statistical collaborator. These authors are world-renowned experts in the field of collecting and understanding the nuances of proxy data and how it links with climate. There are statisticians and groups out there (for example the Past Earth Network) who can help.

Yes, we agree that an experienced Bayesian collaborator would have been useful. However, we hope we have convinced the reviewer that problems were only in the exposition, and have been satisfactorily addressed.

 Models like these are now being studied by statisticians in collaboration with proxy experts. One that is not yet in the palaeoclimate literature (which is perhaps why the authors might have missed it) is that of Ilvonen et al which seems very similar to what the authors are trying to achieve here. A more flexible version can be found in Cahill et al which is in the palaeoclimate literature and uses multiple proxies (forams and d13C) in a Bayesian model for sea level reconstruction. A more recent model is my own Bclim (Parnell et al) which allows for joint inference (i.e. all fossil slices, all taxa, multiple climate variables) to be estimated together, with the aim of reducing uncertainty.

Thank you, we have added these references and accompanying text in the introduction:

"The field of Bayesian palaeoenvironmental statistics is rapidly developing. Recent work includes the development of a pollen-based multinomial regression model that assumes a Gaussian species response, with joint inference across core time-slices (Ilvonen et al 2016) and a foraminifera-based multinomial non-parametric response model that allows for multi-modal and non-Gaussian taxon response curves (Cahill et al 2016). Parnell et al (2016) have published an open source R package Bclim (Parnell et al 2015) that uses pollen response surfaces to generate a series of equally probable joint multivariate climate trajectories."

and

"though we note that Cahill et al (2016) incorporate a second proxy ($\delta^{13}$C) through a prior, assuming a normal likelihood $N(\mu_i, \tau)$ with constant precision $\tau$."

Lastly a note on the figures. Again I

Figure 1 has three lines on three panels

We do not plot individual SRCs, rather we

"Figure 1: Probability-weighted SRC parameters $N_k$ (left-hand axis), $p_k$ (left-hand axis) and $P_k$ (right-hand axis) are plotted for all taxa. Three training-sets are considered: chironomid-based temperature (Matthews-Bird et al., 2016), diatom-based pH (Stevenson et al., 1991) and pollen-based temperature (Bush et al., in prep). The x-axes represent the distinct taxa in the training sets (59 chironomid taxa, 225 diatom taxa and 553 pollen taxa). For each of the three SRC parameters, probability-weighted values are derived for each taxon. These are ordered by increasing value and are plotted sequentially. Horizontal gridlines represent the discrete values allowed within individual SRCs."

Figure 2 contains lines, candlesticks, points, crosses, and three different colours. I've read the caption multiple times, and the text associated with it (which covers 3 different sections), and still cannot work out what's being learnt from this picture.

We have added some additional text that attempts to summarise the main lessons of the picture.

"In summary (see section 3.2), this plot demonstrates: 1) Reconstruction errors of WAPLS1 (circles) and BUMPER (crosses) are similar for all assemblages. 2) Increasing the sampling density of a training set reduces both the reconstruction errors (circles and crosses) and the BUMPER reconstruction uncertainty (solid lines). However, continued benefits beyond a sampling density of ~10 are modest. 3) Reconstructions from assemblages that benefit neither from high richness nor from low tolerances (high species turnover) are associated with significantly greater error and reconstruction uncertainty. We note that the overstatement of BUMPER uncertainty relative to the reconstruction error (solid

lines compared to crosses) is expected for this application to idealised data (see Section 3.2.3).
"

Additionally, we have expanded the text in Section 3.2 in an attempt to better explain these data in detail.

Figure 3 is much more useful, but seems to be hardly mentioned aside from the end of the last paragraph of Section 4.

We now refer to Fig 3 where relevant in the text (sections 4.1, 4.2 and 4.3) and have added some additional text in section 4:

"Notably, Figure 3 illustrates the reconstruction-specific uncertainty $\Delta$ ($\pm 2\Delta$ is plotted), which in general differs significantly from the training-set RMSEP that is usually assumed to describe the uncertainty of WA-PLS approaches."

It was a shame not to see any actual reconstructions of climate over time for any of the sites.

We decided not to include paleo-reconstructions, but rather to focus solely on the transfer function performance, using both idealised and real training sets. BUMPER has been previously applied (albeit without automated priors) to temporal reconstructions. These have been compared with alternative reconstruction methodologies (Holden et al 2008, Matthews-Bird et al 2016) and also with observational data (Holden et al 2008).

References used:

Ilvonen, L., Holmström, L., Seppä, H., and Veski, S. (2016) A Bayesian multinomial regression model for palaeoclimate reconstruction with time uncertainty. Environmetrics, 27: 409–422. doi: 10.1002/env.2393.

Cahill, N., Kemp, A. C., Horton, B. P., & Parnell, A. C. (2016). A Bayesian hierarchical model for reconstructing relative sea level: from raw data to rates of change. Climate of the Past, 12(2), 525-542.

Parnell, A. C., Haslett, J., Sweeney, J., Doan, T. K., Allen, J. R., & Huntley, B. (2016). Joint palaeoclimate reconstruction from pollen data via forward models and climate histories. Quaternary Science Reviews, 151, 111-126.

---

## Referee Report (RR1)

General comments

The clarity of the methods used in the paper is much improved. I have one substantial comment.

Specific comments

On page 4 lines 2-4 you wrote: "and is normalized to 1. It defines the probability density function of the environment, given no prior information and some observed count $y_{k0}$ of taxon $k$."

I wondered in my comment (2 January 2017) whether this normalization was needed and concluded that it was not. This conclusion disregarded the somewhat ad-hoc combination in equation (6). Without the normalization of $L_y$ and $L_p$ (but with normalization of $prob(x)$), we still have the same result for $\eta = 0$ and $\eta = 1$ and the same path of solutions, but for the meaning of $\eta$ would change. In particular, the calculations for $\eta = 0.5$ would give different results (the current result can be obtained with a different value for $\eta$).

The authors can either keep the current normalization (but please deleted the "It defines.." sentence, as it adds nothing) and the ad-hoc combination or change things to a perhaps more defendable combination as follows.

Two models are proposed to infer about $x$. The first model ($M_1$) says that abundance percentages relate to $x$ and the second model ($M_2$) that the presences relates to $x$. So, first a posterior is made on the basis of the first model, say $prob_y(x)$), and then one on the basis of the second model, say $prob_p(x)$. Both probability densities are normalized, of course. Let the prior probability of $M_1$ be $\eta$. Then the final posterior of $x$ is

$$prob(x) = \eta\, prob_y(x) + (1 - \eta)prob_p(x).$$

This construct gives again the same path of solutions as the one with early normalization, but the new construct is a little bit more logical. If desired so, it even allows estimation of $\eta$ on the basis of the posterior probability of $M_1$.

I note for clarity (and you may wish to add it) that equation (4) follows from the law of total probability:

$$prob(y_{k0}|x) = \Sigma_j\, prob(y_{k0}, SRC_{jk}|x) = \Sigma_j\, prob(SRC_{jk})prob(y_{k0}|SRC_{jk}, x)$$

Typo: Equation number (3) is italic.

Cajo ter Braak, Wageningen , 3 January 2017

---

## Author Response (AR2)

**Response to 2nd Review of referee #1 Cajo ter Braak**

Many thanks for this second very thoughtful review.

BUMPER additively combines the two models M1 (percentage) and M2 (p/a) in the construction of individual likelihood functions. We need to normalise them before combining because the likelihoods of the two models have different magnitudes by construction. i.e. prob(presence|x) >> prob(percentage|x). A blended likelihood function without this early normalisation would always be dominated by the p/a model.

The approach suggested by the referee is to construct a posterior from each model, and then to combine these posteriors. We agree that in this case the initial normalisation step would not be required. However, the resulting model is quite different. This difference arises because in the suggested case of combining posteriors (and in contrast to BUMPER) there are no cross-terms between the M1 and M2 likelihood functions i.e. because the two models are derived separately and then additively combined.

We demonstrate this with an example, plotting overleaf the posteriors from the two approaches, considering idealized likelihood functions of two taxa where (at least) one of them is an outlier (i.e. the M1 likelihood functions do not overlap).

In the existing model ("blended likelihoods"), there are two peaks in the posterior. The philosophy of using a blended likelihood function is to prevent an outlying count (e.g. misidentification) from ruling out the "true" reconstruction. This works because the presence likelihood function is generally much broader, so that very little of the reconstruction space is entirely ruled out. (Note that other taxa will generally act together to favour one of these peaks. The consequence is that the outlier will have a weakened effect on the reconstruction.)

In the alternative model, where M1 and M2 are applied separately and the posteriors are combined, the double peak disappears. In effect this model does not consider the possibility that one of the counts is an outlier and this leaves the central region as the most likely reconstruction.

An evaluation of the performance and philosophical differences between these alternative approaches is beyond the scope of this paper, which is focused upon generation of automated priors to allow application to arbitrary proxies, but suggests a potentially interesting avenue for future work.

**Corrections: We have deleted the "It defines…" sentence (P4 L3), replaced it with a note to clarify the need for early normalization, and have added the suggested comment re total law of probability (P3 L39-40)**